# Effects of Aperture Shape on Absorption Property of Acoustic Metamaterial of Parallel-Connection Helmholtz Resonator

**DOI:** 10.3390/ma16041597

**Published:** 2023-02-14

**Authors:** Shaohua Bi, Fei Yang, Shuai Tang, Xinmin Shen, Xiaonan Zhang, Jingwei Zhu, Xiaocui Yang, Wenqiang Peng, Feng Yuan

**Affiliations:** 1Field Engineering College, Army Engineering University of PLA, Nanjing 210007, China; 2Systems Engineering Institute, Academy of Military Sciences, Beijing 100071, China; 3Engineering Training Center, Nanjing Vocational University of Industry Technology, Nanjing 210023, China; 4MIIT Key Laboratory of Multifunctional Lightweight Materials and Structures (MLMS), Nanjing University of Aeronautics and Astronautics, Nanjing 210016, China; 5College of Aerospace Science and Engineering, National University of Defense Technology, Changsha 410073, China; 6Graduate School, Army Engineering University of PLA, Nanjing 210007, China

**Keywords:** sound absorption property, Helmholtz resonator, aperture shape, acoustic metamaterial, acoustic finite element simulation, sound absorption mechanism

## Abstract

A Helmholtz resonator (HR) with an embedded aperture is an effective acoustic metamaterial for noise reduction in the low-frequency range. Its sound absorption property is significantly affected by the aperture shape. Sound absorption properties of HRs with the embedded aperture for various tangent sectional shapes were studied by a two-dimensional acoustic finite element simulation. The sequence of resonance frequency from low to high was olive, common trapeziform, reverse trapeziform, dumbbell and rectangle. Meanwhile, those HRs for various cross-sectional shapes were investigated by a three-dimensional acoustic finite element simulation. The sequence of resonance frequency from low to high were round, regular hexagon, square, regular triangle and regular pentagon. Moreover, the reason for these phenomena was analyzed by the distributions of sound pressure, acoustic velocity and temperature. Furthermore, on the basement of the optimum tangent and cross-sectional shape, the sound absorption property of parallel-connection Helmholtz resonators was optimized. The experimental sample with optimal parameters was fabricated, and its average sound absorption coefficient reached 0.7821 in 500–820 Hz with a limited thickness of 30 mm. The research achievements proved the significance of aperture shape, which provided guidance for the development of sound absorbers in the low-frequency range.

## 1. Introduction

The Helmholtz resonator (HR) is a classical sound absorber which can achieve the desired sound absorption property by tuning its geometric parameters. Many acoustic metamaterials or metastructures based on the HR have been developed for noise reduction [1,2,3,4,5,6,7]. Dogra and Gupta [1] proposed the design of an acoustic metamaterial plate with inbuilt HRs. These types of 3D-printed metamaterial plates could apply in the condition where the high sound transmission loss was desired to a create quieter ambience. Acoustic metamaterial of multiple parallel hexagonal HRs with sub-wavelength dimensions was designed and optimized by Yang et al. [2]. All the actual sound absorption coefficients were above 0.8 in the frequency range of 420–700 Hz with a limited thickness of 40 mm. The modified periodic ducted HR system was proposed by Cai et al. [3], which improved the noise attenuation performance and fully utilized the available space. It was practically used in an actual ventilation ductwork system. Duan et al. [4] proposed the acoustic metamaterials of parallel connection of multiple spiral chambers for low-frequency noise control. It gained an actual sound absorption coefficient above 0.8 in the frequency range of 360–680 Hz and above 0.5 in the frequency range of 350–1600 Hz. The achievement exhibited an extraordinary low-frequency sound absorption property and achieved broadband sound absorption in the low–middle frequency range. By using a microphone and loudspeaker in an HR as the sensor and actuator, respectively, a sweeping HR with time-varying natural frequency was developed by Mao et al. [5]. It eliminated the complexity of changing the physical dimension of the HR during operation. The combined effect of sound absorption at lower frequencies and sound diffusion at higher frequencies was obtained by Herrero-Dura et al. [6] by 2D arrays of HRs. It might play a relevant role in the design of noise reduction systems for different applications. Yang et al. [7] developed the adjustable parallel Helmholtz acoustic metamaterial. With the normal incidence, it could achieve all sound absorption coefficients above 0.9 within the frequency range of 602–1287 Hz and those above 0.85 within the frequency range of 618–1482 Hz.

In order to improve the sound absorption property in the low-frequency range, the common perforated hole is replaced by the embedded aperture. It can shift the resonance frequency to the low-frequency direction [8,9,10,11,12,13,14,15]. The HR with a spiral neck was developed by Shi and Mak [8] to lengthen the neck in a small space. It could achieve fine sound reduction effect within a small space at a low frequency. Xu et al. [9] investigated the acoustic characteristics of HR, which consisted of a pair of cylindrical necks and cavities connected in a series (neck–cavity–neck–cavity). It was further studied by Kim and Selamet [10] in the presence of flow theoretically and experimentally. The sound absorption coefficient of HR was experimentally investigated as a function of the diameter of the neck, the length of the neck and the depth of the cavity by Komkin et al. [11]. It gained the dependence of sound absorption by an HR on its geometric parameters. Langfeldt et al. [12] proposed a new analytical model to calculate the resonance frequency and the input impedance of an HR with multiple necks. It could be applied to design novel HRs or estimate the impact of leaks. Wang and Mak [13] had taken the disorder in a periodic duct-resonator system into consideration. It proved that the disorder in the geometries of resonators provided a useful method to obtain a relatively wide noise attenuation band and to track some narrow noise peaks within it. A tunable piston-cylinder-type HR was designed by Biswas and Agrawal [14] to target a particular frequency. The resonator placed in the cavity volume was further explored to promote the application of the proposed HR. The promising design of the novel HR comprising one small-sized and one large-sized tube in a series was proposed by Chen et al. [15]. It could broaden the bandwidth of absorption and achieve frequency manipulation for the absorption peak. These developed sound absorbers generated from classical HRs promote their practical applications.

Limited by the preparation technique and fabrication accuracy, the shape of the embedded aperture is cylindrical and the cross-sectional shape is circular, although the shape of the chamber can be adjusted for ease of installation [16,17,18,19,20]. Zhang et al. [16] designed kinds of new parallel connections of the HR with the embedded aperture, the working wavelength of which was about 57 times its total thickness of 43 mm. The average sound absorption coefficient was as high as 0.8. Wei et al. [17] investigated the influence of the resonator connection methodology on the frequency response functions of metamaterial beams based on a calculation by the transfer matrix model. The verification was conducted through the finite element simulation. An acoustic metamaterial absorber of parallel connection square HRs was proposed by Wang et al. [18]. It achieved the average actual sound absorption coefficient of 0.9271 in 700–1000 Hz with a total size of 30 mm, 0.9157 in 600–900 Hz with a total size of 40 mm, and 0.9259 in 500–800 Hz with a total size of 50 mm. It was demonstrated by Chiu [19] that the resonator interconnection architectures had a great impact on the global dynamic properties of metamaterials with multiple HRs in the series connection or in the parallel connection. Song et al. [20] constructed an array of HRs with cross-linked polypropylene (IXPP) ferroelectret films on the inner walls of HR cavities. The output power of IXPP films had been significantly improved at multiple frequencies by the series connection of IXPP films.

The development of additive manufacturing technology provides technical support for the design of an aperture with a complex shape, especially the low-force stereolithography, which can fabricate complex structures with high accuracy and no internal residual support by the photopolymer resin [21,22]. Casarini et al. [21] developed acoustic metamaterials based on Helmholtz resonators and capable of attenuating sound up to 30 dB. It was realized by the innovative 3D printing techniques based on stereolithography, and a specific UV-curable resin was utilized. A 2D broadband gradient index acoustic lens was developed by Zigoneanu et al. [22] and fabricated by the rapid prototyping stereolithography. It achieved the broadband performance for frequencies ranging from 1500 Hz to 4500 Hz. Meanwhile, the promotion of the acoustic finite element simulation technology provides an effective and convenient method to investigate the sound absorption performance of acoustic metamaterials based on Helmholtz resonators [23,24,25,26,27,28,29,30]. Cai and Mak [23] validated the noise attenuation capacity of an HR by the finite element method using commercial software (COMSOL Multiphysics). It was only related to the geometries of the neck and the cross-sectional area of the duct. The acoustic simulation software Virtual.Lab was used by Yan et al. [24] to establish the finite element model of a cylindrical shell with an HR. The results indicated that both the noise reduction band and peak amplitude were increased by installing HRs on the cylinder shell. The acoustic performance of the proposed multiple HR arrays system was analyzed numerically by Wu et al. [25,26] with a three-dimensional finite element method simulation. It provided a practical application of hybrid noise control for the ventilation ductwork system and other research areas with respect to HR. Mahesh and Mini [27] investigated the sound absorption characteristics of series and parallel arrangements of both microperforated panels (MPP) and multiple HRs with the inserted neck by full-field finite element simulations. It proved that it could considerably reduce the thickness of the sound absorber. The structural transmission loss of HRs with helical necks was simulated by Bai et al. [28] using finite element analysis software. The overall TL exceeded 30 dB, and the maximum exceeded 60 dB in the target range of 430–2220 Hz. Sun et al. [29] applied the finite element simulation and experimental validation to check the sound absorption performance of multi-frequency perfect sound-absorbing metasurface. The study aimed to obtain alternative solutions to suppress the low-frequency harmonic noise. The three-dimensional finite element method was applied by Cheng et al. [30] to investigate the HR with a horn-shaped neck. It certified that the resonant frequency could be controlled by neck shape and wall thickness. The viscous damping at the neck was the main noise-reduction mechanism of this kind of HR.

Therefore, the effects of aperture shape on the sound absorption performance of HRs were investigated by the acoustic finite element simulation in this study, which included the tangent sectional shape and cross-sectional shape. There were five kinds of tangent sectional shapes studied through the two-dimensional acoustic finite element simulation models. It consisted of a rectangle, common trapeziform, reverse trapeziform, dumbbell and olive. Meanwhile, five kinds of cross-sectional shapes were investigated by the three-dimensional acoustic finite element simulation models. It included a round, regular triangle, square, regular pentagon and regular hexagon. Afterwards, the resonant frequencies and peak sound absorption coefficients for each condition were summarized and studied together. It provided effective guidance for the development of the desired sound absorber with certain constraint conditions. Furthermore, the acoustic metamaterial of multiple parallel connection HRs with the embedded aperture was optimized to gain a satisfactory sound absorption performance for certain application scenarios. It contained a tangent sectional shape of an olive and a cross-sectional shape of a round. The sample was fabricated by low-force stereolithography and detected based on a transfer matrix method to validate its actual sound absorption coefficient. Moreover, the sound absorption mechanism was investigated by the distributions of sound pressure, acoustic velocity and temperature at resonant frequencies. They intuitively exhibited the movement of air in the aperture with high speed and the thermal viscosity effect between air and boundaries.

## 2. Materials and Methods

In this section, the structures of apertures for various shapes were constructed first, and the corresponding acoustic finite element simulation models for the HRs were built subsequently. Afterwards, the sound absorption coefficients of HRs with the aperture for various shapes were obtained in simulation. They provided the effective foundations for the design of actual sound absorber based on the HR and the optimization of its sound absorption performance to satisfy the practical requirement.

### 2.1. Aperture for Various Shapes

The three-dimensional models of apertures for the various shapes were constructed in the 3D modeling software of Solidworks 2019 (Dassault Systèmes SOLIDWORKS Corp., Waltham, MA, USA) in this study, which exhibited the research objects intuitively.

#### 2.1.1. Aperture for Various Tangent Sectional Shapes

The three-dimensional structures of apertures for various tangent sectional shapes investigated in this study are shown in Figure 1. The tangent sectional shapes consist of rectangle, common trapeziform, reverse trapeziform, dumbbell and olive corresponding to Figure 1a–e, respectively. In theory, the tangent sectional shape is arbitrary, but the other complex shapes can be treated as the combination or variation of 5 shapes in Figure 1. Therefore, these five kinds of tangent sectional shapes are selected as the research objects in this study, which provide references for the other kinds of shapes.

Each aperture includes a solid wall and air domain, and the tangent sectional shape of air domain is shown in Figure 2. Here R_u_, R_m_ and R_l_ are the radiuses of upper plane, middle plane and lower plane respectively. It can be observed that the tangent sectional shapes are the rectangle, common trapeziform, reverse trapeziform, dumbbell and olive. That is the basis for naming the HR with embedded aperture for various tangent sectional shapes. The aperture for each tangent sectional shape is rotational symmetric, so the two-dimensional symmetric finite element simulation model can be applied to study the sound absorption performance of HRs with the aperture for various tangent sectional shapes. It is favorable to reduce the simulation time and improve the simulation efficiency simultaneously.

#### 2.1.2. Aperture for Various Cross-Sectional Shapes

The three-dimensional structures of apertures for various cross-sectional shapes investigated in this study are shown in Figure 3. The cross-sectional shapes consist of round, regular triangle, square, regular pentagon and regular hexagon corresponding to Figure 3a–e, respectively. Meanwhile, each aperture includes the solid wall and the air domain, and the cross-sectional shape of air domain are shown in Figure 4. It can be found that the cross-sectional shapes in Figure 4 are rectangle, common trapeziform, reverse trapeziform, dumbbell and olive sequentially. That is the basis for naming the HR with embedded aperture for various cross-sectional shapes.

Supposing the radius of aperture for the cross-sectional shape of round is R, and the side length of regular triangle, square, regular pentagon and regular hexagon are a_rt_, a_s_, a_rp_ and a_rh_ respectively. It can be judged from Figure 4 that the apertures for various cross-sectional shapes are unrotational symmetric except the round shape in Figure 4a. Therefore, the three-dimensional acoustic finite element simulation model is essential for research on the sound absorption performance of HRs with the embedded aperture for various cross-sectional shapes. Meanwhile, the HR with embedded aperture for the rectangle tangent sectional shape in Figure 1a is same with that for round cross-sectional shape in Figure 3a. The two-dimensional model for HR with the embedded aperture in Figure 1a and the three-dimensional model for HR with the embedded aperture in Figure 3a are constructed simultaneously. They can be considered as mutual verification for each other.

### 2.2. Acoustic Finite Element Simulation Method

Acoustic finite element simulation is an effective and convenient method to research the absorption performance of sound-absorbing materials or structures. It is utilized to investigate the sound absorption property of HRs with the embedded aperture for various shapes. Meanwhile, there are two common patterns of acoustic finite element simulation models. One is the two-dimensional symmetric model, which is suitable for rotational symmetric materials or structures, and it can achieve high simulation efficiency. The other is the three-dimensional model, which is appropriate for unrotational symmetric materials or structures. It can gain high simulation accuracy, but its simulation efficiency is low. Thus, taking into consideration the different kinds of aperture for the various shapes, the two-dimensional symmetric models are constructed for the HRs with the embedded aperture for various tangent sectional shapes. The three-dimensional models are built for the HRs with embedded aperture for various cross-sectional shapes.

#### 2.2.1. Two-Dimensional Model of HR with Aperture for Various Tangent Sectional Shapes

The two-dimensional rotational symmetric acoustic finite element models of HRs with the aperture for various tangent sectional shapes are shown in Figure 5. The model includes perfect matching layer (PML), background acoustic field (BAF), the embedded aperture and the rear cavity. It should be noted that the displayed parts in Figure 5 are the air domains.

Supposing the radiuses of upper plane, middle plane and lower plane are R_u_, R_m_ and R_l_, respectively, as shown in Figure 5, the aperture for various tangent sectional shapes can be distinguished by selecting different groups of R_u_, R_m_ and R_l_. The other parameters, such as the length of aperture l, the thickness of cavity T, the diameter of cavity D, the thickness of panel t, the diameter of BAF DD, the thickness of BAF LL and the perforation ratio σ are kept same for the 5 models in Figure 5 and shown in Table 1. The perforation ratio σ is defined as the ratio between the average sectional area of aperture and the sectional area of BAF, as shown in Equation (1). So, the average radius of aperture R_0_ can be derived, as shown in Table 1. The cross-section area of aperture for various tangent sectional shapes in Figure 5 is kept same. Thus, for the tangent sectional shape of rectangle in Figure 5a, it is R_u_ = R_m_ = R_l_ = R_0_ = 2.09 mm. For the tangent sectional shape of common trapeziform in Figure 5b, the R_u_ is set as 1.09 mm. So, it is R_m_ = R_0_ = 2.09 mm and R_l_ = 2 × R_0_ − R_u_ = 3.09 mm. Similarly, for the tangent sectional shape of reverse trapeziform in Figure 5c, the R_u_ is set as 3.09 mm. So, it is R_m_ = R_0_ = 2.09 mm and R_l_ = 2 × R_0_ − R_u_ = 1.09 mm. Meanwhile, for the tangent sectional shape of dumbbell in Figure 5d, the R_u_ is set as 3.09 mm. So, it is R_m_ = 2 × R_0_ − R_u_ = 1.09 mm and R_l_ = R_u_ = 3.09 mm. Moreover, for the tangent sectional shape of olive in Figure 5e, the R_u_ is set as 1.09 mm. So, it is R_m_ = 2 × R_0_ – R_u_ = 3.09 mm and R_l_ = R_u_ = 1.09 mm. Furthermore, the total thickness of investigated HRs can be derived as 30 mm (T + 2 × t = 26 + 2 × 2 = 30). According to the requirement for acoustic finite element simulation with thermoviscous acoustics [7,18], diameter of the PML is equal to that of BAF, and its thickness is 1.5 times that of BAF.
(1)$$\sigma =\frac{{\pi R}_{0}^{2}}{\pi /{4\mathrm{DD}}^{2}}=\frac{{4R}_{0}^{2}}{{\mathrm{DD}}^{2}}$$

The meshed model for the HRs with aperture for various tangent sectional shapes is shown in Figure 6. The major parameters of finite element simulation model in this study are as follows: the size of biggest unit, 0.78 mm; the size of smallest unit, 0.0156 mm; the maximum growth rate of neighboring unit, 1.1; curvature factor, 0.2; the resolution ratio of narrow area, 1; mesh type, free triangle mesh; the layer number of boundary area, 8; the stretching factor of boundary layer, 1.2; the regulatory factor of thickness of boundary layer, 1. Furthermore, the frequency domain solver is chosen for calculation, and the investigated frequency range is set as 200–800 Hz with the interval of 1 Hz. Based on the built finite element models, the sound absorption performance can be gained.

#### 2.2.2. Three-Dimensional Model of HR with Aperture for Various Cross-Sectional Shapes

Schematic diagrams of three-dimensional acoustic finite element model of the HRs with aperture for various cross-sectional shapes is shown in Figure 7. Similar to the two-dimensional model in Figure 5, the model in Figure 7 includes PML, BAF, embedded aperture and the rear cavity as well.

Supposing the radius of aperture for the cross-sectional shape of round in Figure 7a is R, the side lengths of apertures for the cross-sectional shapes of regular triangle, square, regular pentagon and regular hexagon in Figure 7b–e are a_rt_, a_s_, a_rp_ and a_rh_, respectively. In order to keep the volume of air in the aperture equal, the cross-sectional area of various shapes is same, as shown in Equation (2). The area for various cross-sectional shapes of round, regular triangle, square, regular pentagon and regular hexagon can be calculated by Equations (3) to (7), respectively. Thus, the side lengths of apertures for the cross-sectional shapes of regular triangle, square, regular pentagon and regular hexagon are $\sqrt{\frac{4\pi }{\sqrt{3}}}R$, $\sqrt{\pi }R$, $\sqrt{\frac{4\pi }{5\cot\left(\pi /5\right)}}R$ and $\sqrt{\frac{2\pi }{3\sqrt{3}}}R$, respectively. The radius R in Figure 7a is set as 4.18 mm, which is equal to that for two-dimensional rotational symmetric acoustic finite element model of the HR with aperture for tangent sectional shape of rectangle in Figure 5a. It can be utilized to validate the simulation accuracy of two-dimensional model and that of three-dimensional model mutually. The other parameters, such as the length of aperture l, the thickness of cavity T, the diameter of cavity D, the thickness of panel t, the diameter of BAF DD, the thickness of BAF LL and the perforation ratio σ, are kept same with those for the two-dimensional models in Table 1. It aims to compare the sound absorption performances of different HRs simultaneously.
(2)$$S_{r}{=S}_{\mathrm{rt}}{=S}_{s}{=S}_{\mathrm{rp}}{=S}_{\mathrm{rh}}$$
(3)$$S_{r}{=\pi R}^{2}$$
(4)$$S_{\mathrm{rt}}=\frac{\sqrt{3}}{4}a_{\mathrm{rt}}^{2}$$
(5)$$S_{s}{=a}_{s}^{2}$$
(6)Srp=5cot(π5)4arp2
(7)$$S_{\mathrm{rh}}=\frac{3\sqrt{3}}{2}a_{\mathrm{rh}}^{2}$$

Similar to the meshed model of HRs with aperture for various tangent sectional shapes in Figure 6, that of HRs with aperture for various cross-sectional shapes is shown in Figure 8. The major parameters of finite element simulation model in this study are as follows: the size of biggest unit, 1.56 mm; the size of smallest unit, 0.0156 mm; the maximum growth rate of neighboring unit, 1.3; curvature factor, 0.2; the resolution ratio of narrow area, 1; mesh type, free tetrahedron mesh; the layer number of boundary area, 8; the stretching factor of boundary layer, 1.2; the regulatory factor of thickness of boundary layer, 1. Furthermore, the frequency domain solver was chosen for the calculation. The investigated frequency range is set as 200–800 Hz with the interval of 1 Hz. Based on the built finite element models, sound absorption performance can be gained.

### 2.3. Sound Absorption Performance

#### 2.3.1. Absorption Performance of HR with Aperture for Various Tangent Sectional Shapes

The sound absorption coefficients of HRs with aperture for various tangent sectional shapes are shown in Figure 9. It can be intuitively found that the resonance frequencies of HRs with the aperture for tangent sectional shape of rectangle, common trapeziform, reverse trapeziform, dumbbell and olive are 408 Hz, 349 Hz, 355 Hz, 364 Hz and 327 Hz, respectively. The corresponding peak sound absorption coefficients are 0.9966, 0.9565, 0.9598, 0.9942 and 0.9267, successively. The alteration of tangent sectional shape from the traditional rectangle can transform the resonance frequency to the low-frequency direction. The HR with aperture for the tangent sectional shape of olive can obtain the lowest resonance frequency of 327 Hz. The sound absorption performances of HRs with the aperture for various tangent sectional shapes and different parameters are shown in Figure 10. They correspond to the common trapeziform, reverse trapeziform, dumbbell and olive, respectively. The areas of various tangent sectional shapes with different dimensional parameters remain unchanged. It could be judged from Figure 10 that no matter which kind of tangent sectional shape, the resonance frequency shifts to the low-frequency direction along with the decrease in narrowest position of aperture, as shown in Figure 11a. Along with the decrease in narrowest size of aperture from 3.78 mm to 0.98 mm, the resonance frequency decrease from 406 Hz to 240 Hz for common trapeziform, from 406 Hz to 247 Hz for reverse trapeziform, from 408 Hz to 262 Hz for dumbbell and from 400 Hz to 215 Hz for olive, respectively. It can be found that the HR with the aperture for the tangent sectional shape of olive exhibits the best low-frequency sound absorption performance, which is consistent with the results in Figure 9. Meanwhile, it can be judged from the distributions of peak sound absorption coefficients in Figure 10b that there exist optimal geometric parameters for each condition to obtain perfect absorption.

#### 2.3.2. Absorption Performance of HR with Aperture for Various Cross-Sectional Shapes

The sound absorption coefficients of the HRs with an aperture for various cross-sectional shapes are shown in Figure 12. The resonance frequencies for the cross-sectional shapes of round, regular triangle, square, regular pentagon and regular hexagon are 408 Hz, 450 Hz, 423 Hz, 486 Hz and 415 Hz, respectively. The corresponding peak sound absorption coefficients are 0.9988, 0.9692, 0.9989, 0.9621 and 0.9995, successively. It can be found that the HR with the aperture for the cross-sectional shape of round achieves the lowest resonance frequency. The rotational symmetric structure can promote the movement of air in the aperture, which is favorable to result in sound absorption efficiency. Meanwhile, the even shape is conducive to promoting the dissipation of heat transformed from the energy of incident sound wave. That is why most apertures in acoustic metamaterials apply the cross-sectional shapes of round.

#### 2.3.3. Comparative Analysis of Two-Dimensional Model and Three-Dimensional Model

As mentioned above, the HR with aperture for the tangent sectional shape of rectangle in Figure 5a is equal to that for the cross-sectional shape of round in Figure 7a, and the geometric parameters of them two are same as well. The comparison of sound absorption performances obtained by the two-dimensional model and those achieved by the three-dimensional model is shown in Figure 13. It can be observed that they are almost the same, especially in the range near the resonance frequency. The deviations that exist in the range far from the resonance frequency (such as the 200–350 Hz or 500–800 Hz) have relatively little impact because the corresponding sound absorption coefficients are basically below 0.1. They have almost no contribution to the sound absorption performance of acoustic metamaterials. Therefore, the two-dimensional model is utilized to select the initial values of geometric parameters, and the three-dimensional model is used to identify the optimal parameters accurately, which can play into their respective advantages. The aim is to improve the calculation and optimization efficiency for the development of acoustic metamaterial with certain constraint conditions [2,18].

## 3. Designs and Optimizations

Normally speaking, the length of embedded aperture in the rear chamber cannot be too long; otherwise, sound absorption performance of the HR will deteriorate [2,18]. Thus, in order to improve the sound absorption performance of sound absorber with a limited total thickness, the change in shape of the embedded aperture is also a feasible and effective method to obtain better sound absorption performance in the low-frequency direction. It can be judged from Section 2.3 that relative to other shapes, the aperture for the tangent sectional shape of olive and that for the cross-sectional shape of round is favorable to achieve better sound absorption performance in the low-frequency range with the same length of aperture. Thus, the multiple parallel connection of HRs with aperture for the tangent sectional shape of olive and the cross-sectional shape of round is proposed, which aims to obtain excellent sound absorption performance in the low-frequency range.

### 3.1. Structural Design

Three-dimensional acoustic finite element simulation model for the multiple parallel connection HRs with embedded aperture is shown in Figure 14. It consists of a tangent sectional shape of olive and a cross-sectional shape of round. They are similar with the construction of acoustic finite element models in Figure 7 and Figure 8. For the sake of following discussion, the HRs are labeled as H01 to H16, as shown in Figure 14b. A single HR is shown in Figure 14c, which consists of an embedded aperture and a rear cavity. The thicknesses of front panel, back panel and side wall are set as 2 mm. The major parameters for this simulation model are as follows: the size of biggest unit, 2 mm; the size of smallest unit, 0.02 mm; the maximum growth rate of neighboring unit, 1.3; curvature factor, 0.2; the resolution ratio of narrow area, 1; mesh type, free tetrahedron mesh; the layer number of boundary area, 8; the stretching factor of boundary layer, 1.2; the regulatory factor of thickness of boundary layer, 1. Furthermore, the frequency domain solver was chosen for the calculation, and the investigated frequency range was selected at 200–1600 Hz with an interval of 2 Hz. The structural parameters of apertures are adjusted to gain the desired sound absorption performance, which includes the radius of upper plane R_u_, the radius of middle plane R_m_ and the length of aperture L.

Meanwhile, the normal multiple parallel connection of the HRs with the embedded cylindrical aperture is investigated as well. They can be treated as the contrasts. The radiuses of apertures are set as equal to the radius of upper plane R_u_, the radius of middle plane R_m_ and the average of them two (R_u_ + R_m_)/2, respectively, as shown in Figure 15. The length of each embedded aperture is kept equal to that in Figure 14.

### 3.2. Parameter Optimization

In order to decrease the parameter number and increase the optimization efficiency, the 16 HRs in Figure 14b are divided into 4 groups, as H01–H04, H05–H08, H09–H12 and H13–H16. For each group, the radius of upper plane R_u_ and that of middle plane R_m_ are equal, and the length of each aperture is altered to broaden the effective frequency bandwidth. According to the exhibited sound absorption properties of the HR with olive aperture in Figure 10d, the R_u_ and R_m_ are set and summarized in Table 2. Moreover, the length of aperture for each HR L_i_ (i = 1,2,…,16) is the parameter to be optimized for a certain requirement. Taking a given workshop, for example, the required sound absorption coefficients were larger than 0.7 in 500–820 Hz, and the available space was 30 mm. Normally speaking, the actual sound absorption coefficients were smaller than the simulation data, and the effective sound absorption band would decrease. Therefore, the constraint condition was turned to the desired sound absorption band of 460–860 Hz and all the sound absorption coefficients above 0.7. According to the constraint conditions, the parameters L_i_ were optimized by the cuckoo search algorithm [31,32,33,34], and the results were summarized in Table 2.

### 3.3. Sample Fabrication

The multiple parallel connection HRs with the embedded aperture is fabricated by the low-force stereolithography 3D printer Form3 (Formlabs Inc., Summerville, MA, USA) [35], as shown in Figure 16. It consists of a tangent sectional shape of olive and a cross-sectional shape of round. The 3D model of acoustic metamaterial is built in the 3D modeling software Solidworks (Dassault Systèmes SOLIDWORKS Corp., Waltham, MA, USA), as shown in Figure 16a. Afterwards, through the format conversion of model and the import of it into Form3, the desired sample is fabricated, as shown in Figure 16c. When the fabrication of sample is finished, it is further cleaned by FormlabsForm Wash (Formlabs Inc., Summerville, MA, USA) to remove the residual liquid resin and irradiated for solidification by FormlabsForm Cure (Formlabs Inc., Summerville, MA, USA). Meanwhile, the three samples for contrasts are prepared as well in a similar way.

### 3.4. Sound Absorption Coefficient Measurement

The actual sound absorption coefficients of these fabricated samples are measured by an AWA6290T transfer function impedance tube detector (Hangzhou Aihua Instruments Co., Ltd., Hangzhou, China). It accords with the national standard of GB/T 18696.2–2002 (ISO 10534–2:1998) “Acoustics–Determination of sound absorption coefficient and impedance in impedance tubes–part 2: Transfer function method” [36,37,38]. The schematic diagram and actual image are shown in Figure 17a,b, respectively [2,4,7,18]. The detected acoustic metamaterials of multiple parallel connection HRs are fixed in the end of impedance tube and held by the cavity adjuster. The sound source is installed on the other end of impedance tube. The white noise signal is generated from the noise generator and amplified by the power amplifier, and it is further input to the sound source for transmission to the incident wave. The reflected wave is detected by 2 microphones fixed on the impedance tube with a distance of 70 mm, and the received signals are further dealt with by the dynamic signal analyzer and the software in the workstation. Meanwhile, the distance between microphone 2 and front surface of the detected acoustic metamaterial is set as 170 mm according to the requirement of utilized AWA6290T detector. In this way, the actual sound absorption coefficients of detected acoustic metamaterials in the frequency range of 200–1600 Hz can be obtained.

## 4. Results and Discussions

The sound absorption property of the proposed acoustic metamaterial of multiple parallel connection HRs with an embedded aperture that consists of the tangent sectional shape of an olive and the cross-sectional shape of a round is analyzed. Its sound absorption mechanism is discussed as well based on the distributions of sound pressure.

### 4.1. Sound Absorption Performance

#### 4.1.1. Comparisons of Simulation Results and Experimental Results

The comparisons of sound absorption coefficients obtained in the simulation with those achieved in the impedance tube measurement are shown in Figure 18. It can be found that the consistencies between the simulation data and experimental data for each sample are satisfactory. The major reason for the derivation is that there exist fabrication errors for the parameters of prepared samples, and the acoustic finite element model was built completely consistent with the desired theoretical parameters [39,40,41,42]. Moreover, it can be judged from Figure 18a that the theoretical sound absorption coefficients in 460–860 Hz are larger than 0.7, which meets the simulation target and proves the effectiveness of the cuckoo search optimization method. Meanwhile, the actual sound absorption coefficients of the multiple parallel connection HRs with the olive aperture are larger than 0.7 in the target frequency range of 500–820 Hz. This satisfies the requirement of noise reduction for the given workshop. The actual minimal and maximal sound absorption coefficient in this region is 0.7362 and 0.8199, respectively, and the average value is 0.7821. This exhibits a satisfactory and relative homogeneous sound absorption performance.

#### 4.1.2. Comparisons of Absorption Properties of the Proposed Acoustic Metamaterial with Those of Multiple Parallel Connection HRs with Normal Cylindrical Aperture

The comparisons of actual sound absorption performances of the proposed acoustic metamaterial with those of multiple parallel connection HRs with the normal cylindrical apertures are shown in Figure 19. Along with the radius increasing from R_u_ to R_m_, the sound absorption coefficients of multiple parallel connection HRs with the normal cylindrical aperture shift to the high frequency direction. Meanwhile, judging from the sound absorption performance of the proposed acoustic metamaterial and that of multiple parallel connection HRs with the normal cylindrical aperture of R_u_, it can be found that the length of the aperture with a minimum radius would affect absorption bandwidth and efficiency as well. This indicates that the sound absorption property of multiple parallel connection HRs with olive aperture can be further improved in a low-frequency region by using multiple olives.

Moreover, it can be found that the actual average sound absorption coefficient for the optimal sample is 0.7821 in the frequency range of 500–820 Hz. The total thickness of the sound absorber is 30 mm, which is approximately 1/22 of the maximum wavelength λ_max_ (λ_max_ = c/f_min_ = 340/500 m = 0.68 m = 680 mm). The sound absorption efficiency is quite high for a sound absorber with a broad absorption bandwidth. In contrast, the total thickness was near 1/19 of the maximum wavelength λ_max_ for the achievement in reference [2]. Meanwhile, the total thickness was near 1/12, 1/16 and 1/17 of the maximum wavelength λ_max_ for the achievements in references [4,18,43], respectively. Therefore, it can be found that the proposed acoustic metamaterial of multiple parallel connection HRs with embedded aperture that consist of a tangent sectional shape of olive and a cross-sectional shape of round is efficient.

### 4.2. Sound Absorption Mechanism

#### 4.2.1. Absorption Mechanism of HR with Aperture for Various Tangent Sectional Shapes

Based on the constructed acoustic finite element simulation model in Figure 5 and Figure 6, the sound absorption mechanism of the HR with an aperture for various tangent sectional shapes is analyzed by investigating the distributions of temperature, acoustic velocity and acoustic pressure for the corresponding resonance frequencies [2,4,6,7,18], as shown in Figure 20, Figure 21 and Figure 22, respectively. In order to make the results more intuitive, a two-dimensional rotation of the plane finite element model from −90° to 135° is conducted. It can be found from the distribution of temperature in Figure 20 that the main temperature rise occurs at the boundary area of the rear chamber, and that of the embedded aperture also shows some temperature rise. Meanwhile, it can be observed from the distribution of acoustic velocity in Figure 21 that except for the HR with the aperture for the tangent sectional shape of a rectangle, all the other 4 conditions show remarkable velocity improvement in the narrowest position of the embedded aperture. Moreover, it can be seen from the distribution of acoustic pressure in Figure 22 that the air in the rear chamber shows high pressure. The selected acoustic type for BAF in this study is the plane wave with the amplitude of 1 Pa and direction of [0, –1]. The PML can completely absorb all the transmitted sound wave, so its acoustic pressure is 0. The pressure in the embedded aperture gradually transits from 1 Pa in the BAF to a large level in the rear chamber. Since the sound absorption coefficient is derived from acoustic pressures in the embedded aperture and those in the rear chamber, the larger acoustic pressures always indicate a higher sound absorption coefficient. It can be judged from the contrast of the distribution of the acoustic pressure in Figure 22 with that of the simulated sound absorption coefficient in Figure 9. Furthermore, the acoustic velocity for the HR with the aperture for the tangent sectional shape of a rectangle is obviously smaller than that for the other four conditions, especially relative to the HR with the aperture for the tangent sectional shape of an olive. It has two narrow positions and generates two acoustic velocity peaks. The distributions of the acoustic velocity are major factors in deciding the resonance frequency, which are consistent with the simulation results in Figure 9.

#### 4.2.2. Absorption Mechanism of HR with Aperture for Various Cross-Sectional Shapes

Similarly, according to the constructed acoustic finite element simulation models in Figure 7 and Figure 8, the sound absorption mechanism of the HR with the aperture for various cross-sectional shapes are analyzed as well by investigating the distributions of the temperature, acoustic velocity and acoustic pressure for these corresponding resonance frequencies [2,4,6,7,18], as shown in Figure 23, Figure 24 and Figure 25, respectively. The exhibited characteristics are mainly consistent with those in Figure 20, Figure 21 and Figure 22. The main conversion from sound energy to heat energy occurs at the boundary area of the rear chamber (as shown in Figure 23). The high frequency movement of air in the embedded aperture (as shown in Figure 24) results in the increase in acoustic pressure in the rear chamber (as shown in Figure 25). It can be observed from Figure 24 that the acoustic velocity for the HR with the aperture for a round is larger than those for the HRs with other apertures, which can realize resonance in the lower sound frequency. Meanwhile, it can be observed that the sort of acoustic velocity for the HR with various cross-sectional shapes from large to small in Figure 12 is round, regular hexagon, square, regular triangle and regular pentagon. The results are consistent with the simulation results in Figure 12. These exhibited characteristics are consistent with the normal sound absorption mechanism of multiple parallel connection of HRs [2,7,18]. They can certify the correctness of this research about the effects of tangent- and cross-sectional shapes of the aperture on the sound absorption performance of HRs.

#### 4.2.3. Absorption Mechanism of Multiple Parallel Connection of HRs

Distributions of the acoustic pressure for the proposed multiple parallel connection HRs with embedded apertures are shown in Figure 26. It consists of the tangent sectional shape of an olive and the cross-sectional shape of a round. They are the normal method to exhibit the sound absorption mechanism. The sound absorption coefficients obtained by the acoustic finite element simulation in Figure 18a are taken into consideration. The investigated frequency points are 400 Hz, 420 Hz, 450 Hz, 500 Hz, 550 Hz, 600 Hz, 650 Hz, 700 Hz, 750 Hz, 800 Hz, 850 Hz and 900 Hz, respectively. It can be judged from Figure 18a that the effective sound absorption band is 450–850 Hz with all the sound absorption coefficient above 0.7. According to the serial number for each HR in Figure 14b, it can be found from Figure 26c–k that the sound absorption effects for the frequency points from 450 Hz to 850 Hz with the interval of 50 Hz are obtained by the H12, H11 and H10, H08, H07 and H06, H05 and H04, H04 and H03, H03 and H02, H02 and H01, and H01, respectively. Meanwhile, it can be observed that the HRs with the serial number of H13, H14, H15 and H16 are effective only when the frequency is lower than 450 Hz, as shown in Figure 26a,b. The achievement of effective sound absorption performance in the low-frequency area is difficult relative to that in the high frequency region. It indicates that the arrangement of more HRs in the low-frequency range is essential. Otherwise, the first resonance frequency cannot be generated, and the obtained effective frequency range will shift to the high frequency direction.

## 5. Conclusions

The effects of tangent- and cross-sectional shapes on the sound absorption performance of the acoustic metamaterial of multiple parallel connection HRs is researched and validated in this study. The sound absorption mechanism is investigated and exhibited by the acoustic finite element simulation. The major achievements obtained in this study are as follows:(1)The sound absorption coefficients of HRs with apertures for various tangent sectional shapes are investigated by the two-dimensional rotational symmetric acoustic finite element model. It consists of the rectangle, common trapeziform, reverse trapeziform, dumbbell and olive. It can be concluded that the alteration of the tangent sectional shape from the traditional rectangle can shift the resonance frequency to the low-frequency direction. The HR with the aperture for the tangent sectional shape of the olive can obtain the lowest resonance frequency of 327 Hz.(2)The sound absorption properties of HRs with apertures for various cross-sectional shapes is studied by the three-dimensional acoustic finite element model. It includes the round, regular triangle, square, regular pentagon and regular hexagon. It has been proved that the HR with the aperture for the cross-sectional shape of the round can achieve the lowest resonance frequency. That is why most apertures in the acoustic metamaterials apply the cross-sectional shapes of a round.(3)The sound absorption performance gained by the two-dimensional model is consistent with that achieved by the three-dimensional model with the same geometric parameters. It indicates that the two-dimensional model can be used to determine the initial values of geometric parameters, and the three-dimensional model can be applied to identify the optimal parameters accurately. It is favorable to improve the calculation and optimization efficiency for the development of some acoustic metamaterial with certain constraint conditions.(4)The acoustic metamaterial of multiple parallel connection of the HRs with apertures for the tangent sectional shape of the olive and the cross-sectional shape of the round is optimized by the cuckoo search algorithm. The samples are prepared by the low-force stereolithography 3D printer Form3 and detected by the AWA6290T transfer function impedance tube detector. The actual average sound absorption coefficient of 0.7821 in the frequency range of 500–820 Hz is obtained with a limited thickness of 30 mm. It exhibits a satisfactory and relatively homogeneous sound absorption performance to meet the requirement of noise reduction for the given application scenario.(5)The sound absorption mechanism of HRs with apertures for various tangent sectional shapes and that of HRs with apertures for various cross-sectional shapes are investigated by the distributions of sound pressure, acoustic velocity and temperature at these resonance frequencies. The achieved characteristics are consistent with the normal sound absorption mechanism for the multiple parallel connection of HRs. They can certify the correctness of this study about the effects of tangent- and cross-sectional shapes of the aperture on the sound absorption performance of HRs.

## Figures and Tables

**Figure 1 materials-16-01597-f001:**
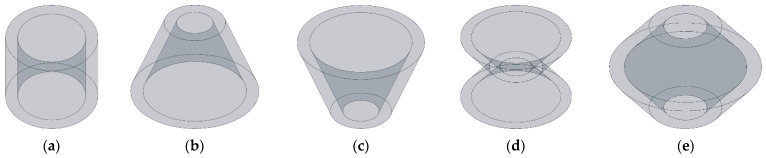
The three-dimensional structures of apertures for various tangent sectional shapes. (**a**) Rectangle; (**b**) common trapeziform; (**c**) reverse trapeziform; (**d**) dumbbell; and (**e**) olive.

**Figure 2 materials-16-01597-f002:**
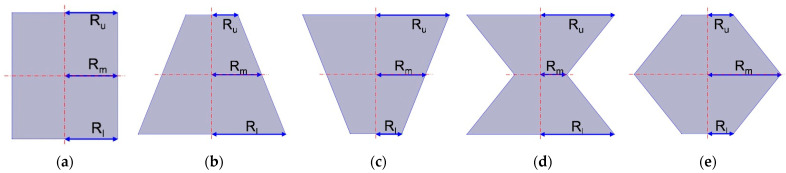
The tangent sectional shape of air domain in aperture for various tangent sectional shapes. (**a**) Rectangle; (**b**) common trapeziform; (**c**) reverse trapeziform; (**d**) dumbbell; and (**e**) olive.

**Figure 3 materials-16-01597-f003:**
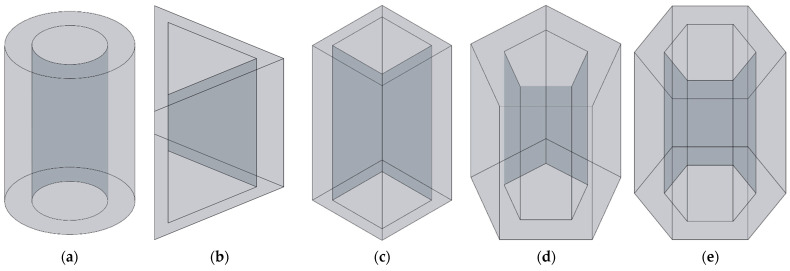
The three-dimensional structures of apertures for various cross-sectional shapes. (**a**) Round; (**b**) regular triangle; (**c**) square; (**d**) regular pentagon; and (**e**) regular hexagon.

**Figure 4 materials-16-01597-f004:**
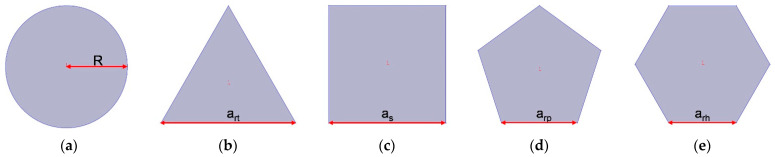
The cross-sectional shape of air domain in the aperture for various cross-sectional shapes. (**a**) Round; (**b**) regular triangle; (**c**) square; (**d**) regular pentagon; and (**e**) regular hexagon.

**Figure 5 materials-16-01597-f005:**
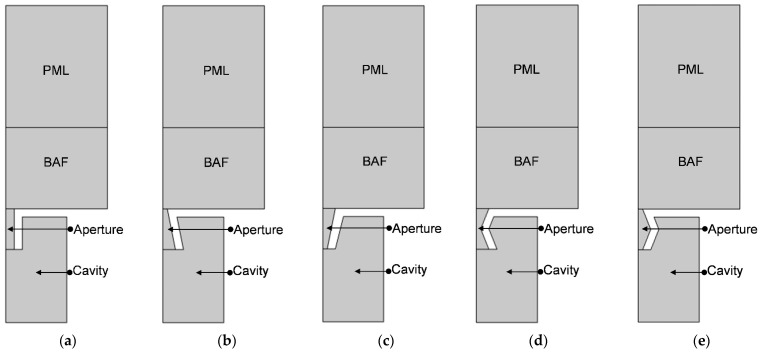
The two-dimensional rotational symmetric acoustic finite element model of HRs with the aperture for various tangent sectional shapes. (**a**) Rectangle; (**b**) common trapeziform; (**c**) reverse trapeziform; (**d**) dumbbell; and (**e**) olive.

**Figure 6 materials-16-01597-f006:**
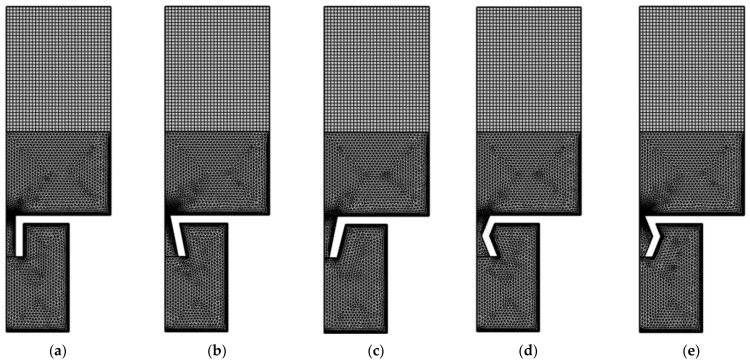
The meshed model for the HRs with an aperture for various tangent sectional shapes. (**a**) Rectangle; (**b**) common trapeziform; (**c**) reverse trapeziform; (**d**) dumbbell; and (**e**) olive.

**Figure 7 materials-16-01597-f007:**
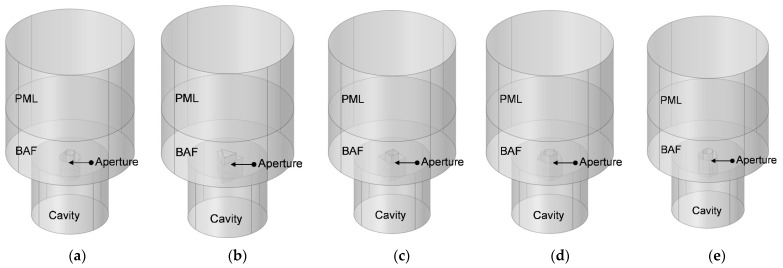
The three-dimensional finite element model for HRs with aperture for various cross-sectional shapes. (**a**) Round; (**b**) regular triangle; (**c**) square; (**d**) regular pentagon; and (**e**) regular hexagon.

**Figure 8 materials-16-01597-f008:**

The meshed model for HRs with aperture for various tangent sectional shapes. (**a**) Round; (**b**) regular triangle; (**c**) square; (**d**) regular pentagon; and (**e**) regular hexagon.

**Figure 9 materials-16-01597-f009:**
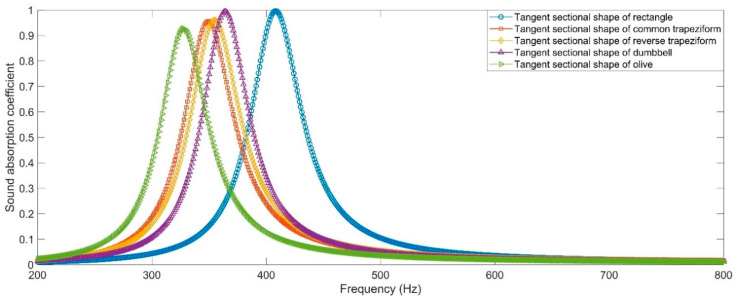
The sound absorption performance of HRs with the aperture for various tangent sectional shapes.

**Figure 10 materials-16-01597-f010:**
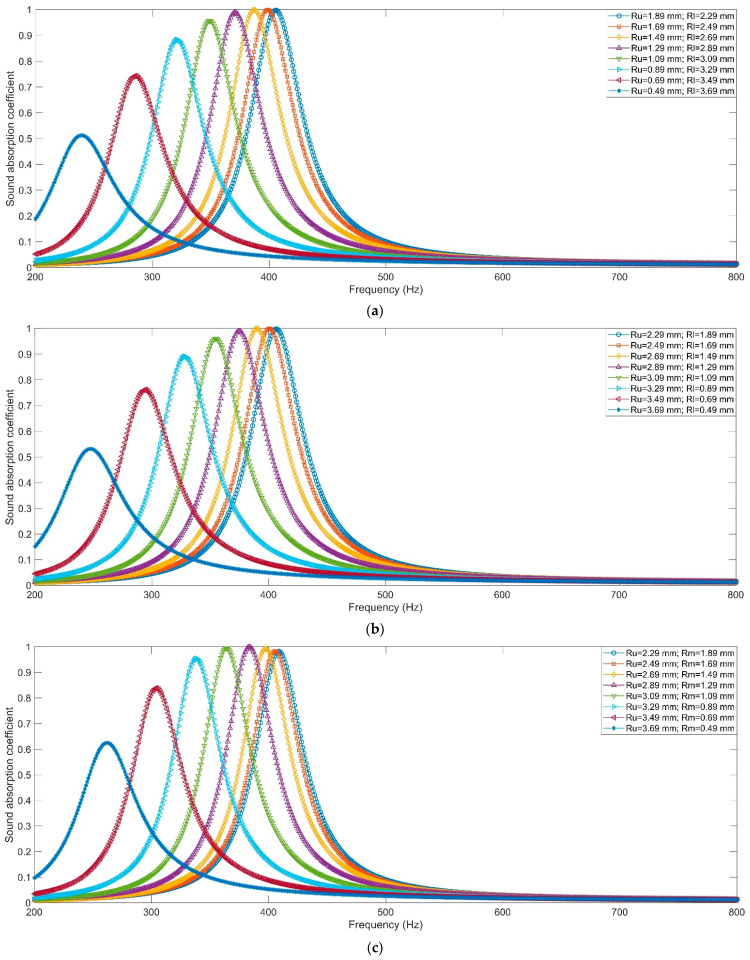
The sound absorption property of HR with the aperture for various tangent sectional shapes and different parameters. (**a**) Common trapeziform; (**b**) reverse trapeziform; (**c**) dumbbell; and (**d**) olive.

**Figure 11 materials-16-01597-f011:**
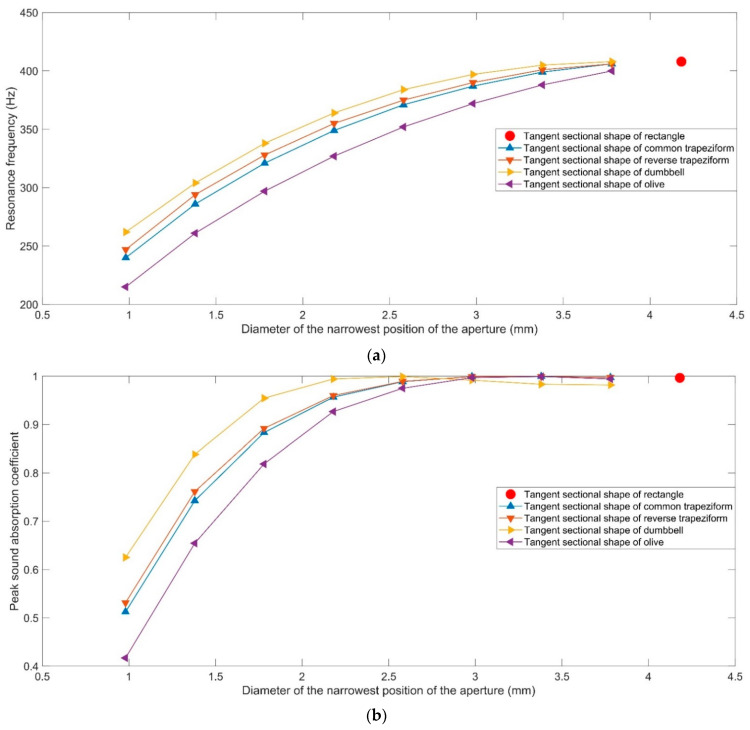
Comparisons of sound absorption performance of HRs with aperture for various tangent sectional shapes. (**a**) Resonance frequency; and (**b**) peak sound absorption coefficient.

**Figure 12 materials-16-01597-f012:**
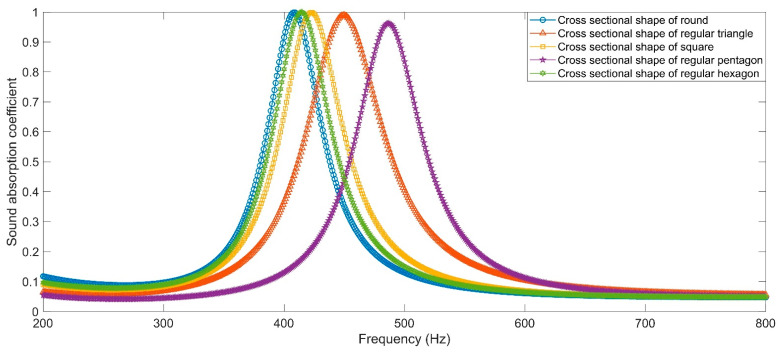
Sound absorption performance of HRs with aperture for various cross sectional shape.

**Figure 13 materials-16-01597-f013:**
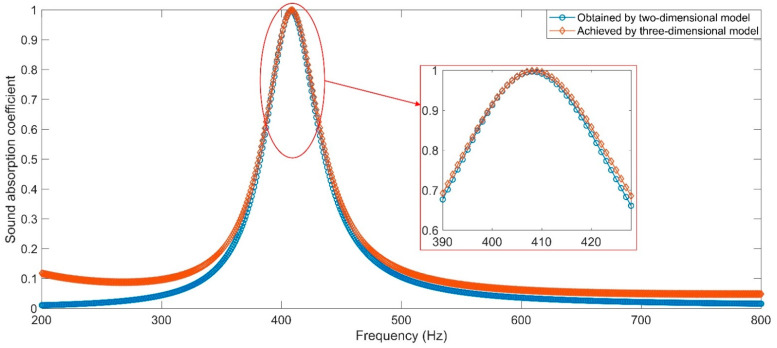
The comparison of sound absorption performances obtained by two-dimensional model and those achieved by three-dimensional model.

**Figure 14 materials-16-01597-f014:**
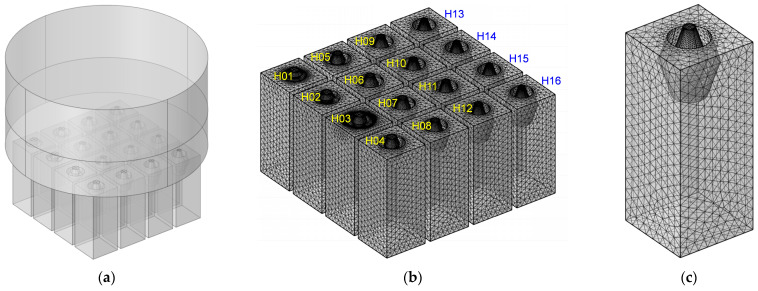
Three-dimensional finite element simulation model for the multiple parallel connection HRs with embedded aperture which consist of a tangent sectional shape of olive and a cross-sectional shape of round. (**a**) The acoustic finite element simulation model; (**b**) the meshed model for multiple parallel connection HRs; and (**c**) the single HR.

**Figure 15 materials-16-01597-f015:**
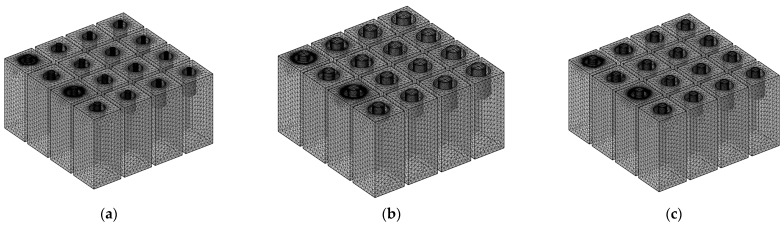
Three-dimensional finite element simulation model for the multiple parallel connection HRs with normal cylindrical aperture. (**a**) The radius of aperture is equal to upper plane R_u_; (**b**) the radius of aperture is equal to middle plane R_m_; and (**c**) the radius of aperture is equal to (R_u_ + R_m_)/2.

**Figure 16 materials-16-01597-f016:**
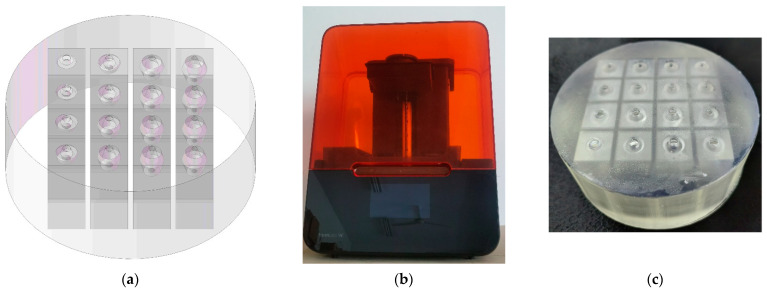
The fabrication of proposed experimental sample. (**a**) The three-dimensional structure of sample; (**b**) the low-force stereolithography 3D printer; and (**c**) the fabricated sample.

**Figure 17 materials-16-01597-f017:**
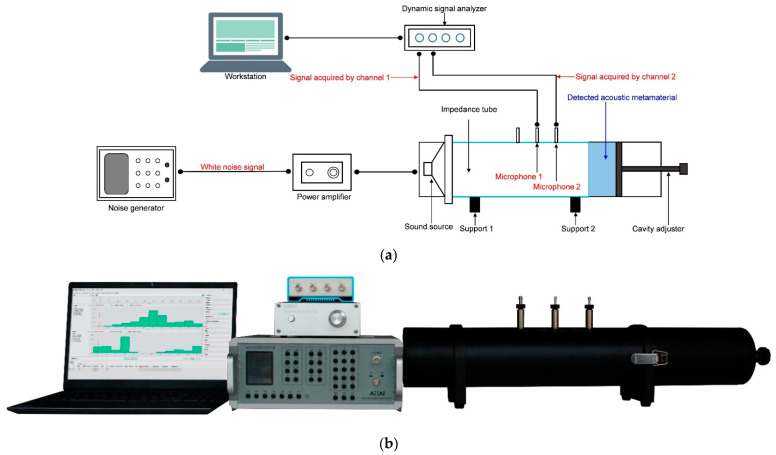
The sound absorption coefficient measurement. (**a**) Schematic diagram of measurement process; and (**b**) the actual image of AWA6290T transfer functional impedance detector.

**Figure 18 materials-16-01597-f018:**
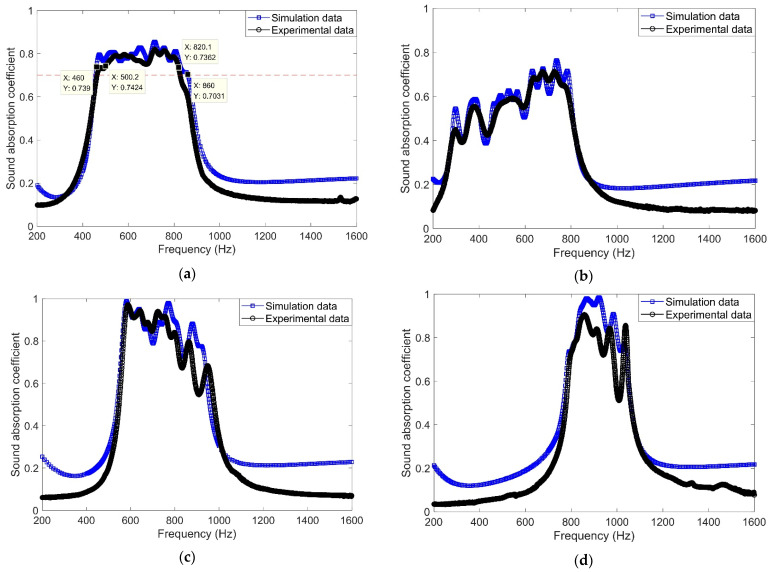
The comparison of sound absorption coefficients obtained in the simulation with those achieved in impedance tube measurement. (**a**) The proposed acoustic metamaterial in this research; (**b**) the multiple parallel connection HRs with normal cylindrical aperture for the radius of R_u_; (**c**) the multiple parallel connection HRs with normal cylindrical aperture for the radius of (R_u_ + R_m_)/2; and (**d**) the multiple parallel connection HRs with normal cylindrical aperture for the radius of R_m_.

**Figure 19 materials-16-01597-f019:**
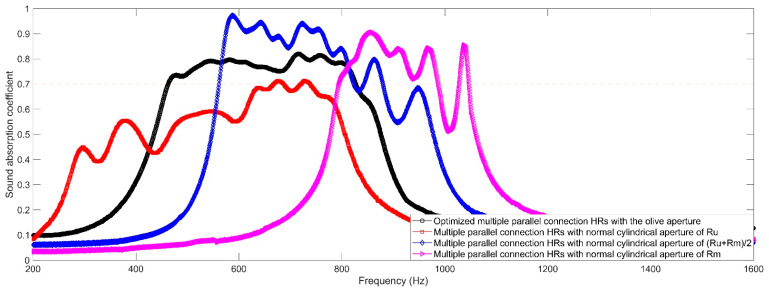
Comparison of sound absorption performances of the proposed acoustic metamaterial with those of the multiple parallel connection HRs with normal cylindrical aperture.

**Figure 20 materials-16-01597-f020:**
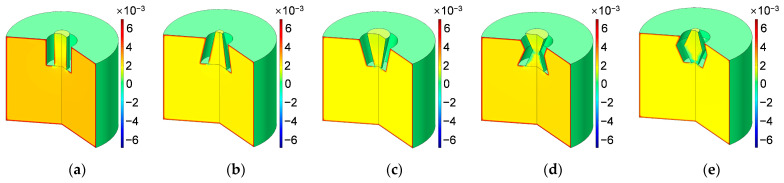
Distribution of temperature for the HR with aperture for various tangent sectional shapes. (**a**) Rectangle; (**b**) common trapeziform; (**c**) reverse trapeziform; (**d**) dumbbell; and (**e**) olive.

**Figure 21 materials-16-01597-f021:**
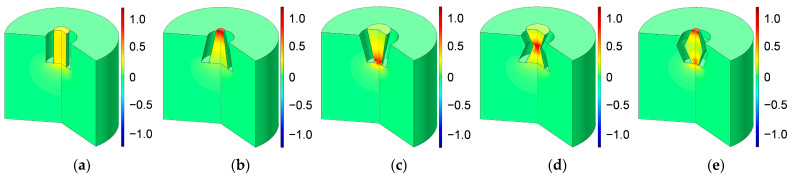
Distribution of acoustic velocity for the HR with aperture for various tangent sectional shapes. (**a**) Rectangle; (**b**) common trapeziform; (**c**) reverse trapeziform; (**d**) dumbbell; and (**e**) olive.

**Figure 22 materials-16-01597-f022:**
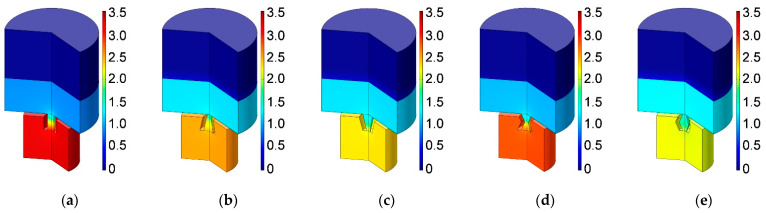
Distribution of acoustic pressure for the HR with aperture for various tangent sectional shapes. (**a**) Rectangle; (**b**) common trapeziform; (**c**) reverse trapeziform; (**d**) dumbbell; and (**e**) olive.

**Figure 23 materials-16-01597-f023:**

Distribution of temperature for the HR with aperture for various cross-sectional shapes. (**a**) Round; (**b**) regular triangle; (**c**) square; (**d**) regular pentagon; and (**e**) regular hexagon.

**Figure 24 materials-16-01597-f024:**

Distribution of acoustic velocity for the HR with the aperture for various cross-sectional shapes. (**a**) Round; (**b**) regular triangle; (**c**) square; (**d**) regular pentagon; and (**e**) regular hexagon.

**Figure 25 materials-16-01597-f025:**
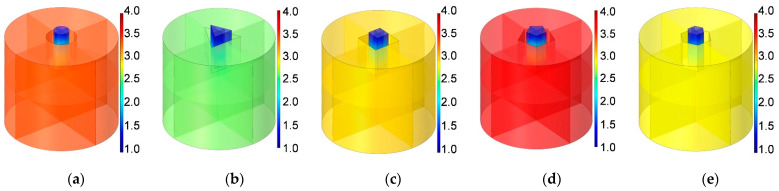
Distribution of acoustic pressure for the HR with the aperture for various cross-sectional shapes. (**a**) Round; (**b**) regular triangle; (**c**) square; (**d**) regular pentagon; and (**e**) regular hexagon.

**Figure 26 materials-16-01597-f026:**
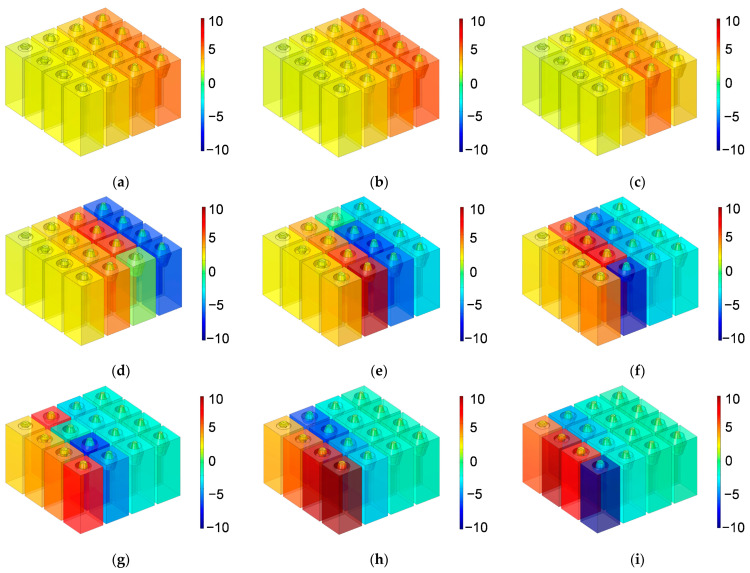
Distributions of acoustic pressure for multiple parallel connection HRs with embedded aperture, which consist of tangent sectional shape of olive and cross-sectional shape of round. (**a**) 400 Hz; (**b**) 420 Hz; (**c**) 450 Hz; (**d**) 500 Hz; (**e**) 550 Hz; (**f**) 600 Hz; (**g**) 650 Hz; (**h**) 700 Hz; (**i**) 750 Hz; (**j**) 800 Hz; (**k**) 850 Hz; and (**l**) 900 Hz.

**Table 1 materials-16-01597-t001:** The summarized parameters for two-dimensional rotational symmetric acoustic finite element models.

Parameters	Parameters of Model for the HRs with Aperture for Various Tangent Sectional Shapes				
	Rectangle	Common Trapeziform	Reverse Trapeziform	Dumbbell	Olive
Radius of the upper plane R_u_	2.09 mm	1.09 mm	3.09 mm	3.09 mm	1.09 mm
Radius of the middle plane R_m_	2.09 mm	2.09 mm	2.09 mm	1.09 mm	3.09 mm
Radius of the lower plane R_l_	2.09 mm	3.09 mm	1.09 mm	3.09 mm	1.09 mm
Length of the aperture l	10 mm				
Thickness of the cavity T	26 mm				
Diameter of the cavity D	30 mm				
Thickness of the panel t	2 mm				
Diameter of the BAF DD	50 mm				
Thickness of the BAF LL	20 mm				
Perforation ratio σ	0.7%				

**Table 2 materials-16-01597-t002:** The optimal parameters of multiple parallel connection HRs with the olive apertures (unit: mm).

	R_u_	R_m_	R_l_	L		R_u_	R_m_	R_l_	L		R_u_	R_m_	R_l_	L		R_u_	R_m_	R_l_	L
H01	1.69	2.49	1.69	2.3	H05	1.49	2.69	1.49	5.2	H09	1.29	2.89	1.29	9.4	H13	1.09	3.09	1.09	12.4
H02	1.69	2.49	1.69	3.1	H06	1.49	2.69	1.49	6.4	H10	1.29	2.89	1.29	10.2	H14	1.09	3.09	1.09	12.5
H03	1.69	2.49	1.69	4.2	H07	1.49	2.69	1.49	7.8	H11	1.29	2.89	1.29	11.4	H15	1.09	3.09	1.09	12.6
H04	1.69	2.49	1.69	5.3	H08	1.49	2.69	1.49	9.2	H12	1.29	2.89	1.29	12.4	H16	1.09	3.09	1.09	12.7

## Data Availability

The data that support the findings of this study are available from the corresponding author upon reasonable request.

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
