# Peer review of "Effects of Aperture Shape on Absorption Property of Acoustic Metamaterial of Parallel-Connection Helmholtz Resonator"

_materials, 2023, doi:10.3390/ma16041597_

Round 1

Reviewer 1 Report

In my judgment, the paper can be published in this special issue. However, it has some issues that must be carefully addressed before considering for publication  

1. The language should be improved. The text is full of vague and long sentences such as: With same other geometric parameters, HRs with the embedded aperture for tangent sectional shape of rectangle, common trapeziform, reverse trapeziform, dumbbell and olive gained the sound absorption peak at the frequency of 408 Hz, 349 Hz, 355 Hz, 364 Hz and 328 Hz respectively through two–dimensional acoustic finite element simulation.

2.The results should be compared with the state-of-the-art related works.

3.Why HR with the aperture for the cross sectional shape of round achieves the lowest resonance frequency? Please explain in detail. 

Author Response

Response to reviewer 1

General Comment: In my judgment, the paper can be published in this special issue. However, it has some issues that must be carefully addressed before considering for publication.

Response:

Thank you very much for your kind review to our manuscript and positive assessment to our research. We have revised the manuscript carefully according to your and the other reviewers’ comments. The responses to your comments are as follows.

  1. The language should be improved. The text is full of vague and long sentences such as: With same other geometric parameters, HRs with the embedded aperture for tangent sectional shape of rectangle, common trapeziform, reverse trapeziform, dumbbell and olive gained the sound absorption peak at the frequency of 408 Hz, 349 Hz, 355 Hz, 364 Hz and 328 Hz respectively through two–dimensional acoustic finite element simulation.

Response:

Thank you very much for your helpful suggestion. According to your and other reviewers’ comment, the language of whole manuscript was polished by a native speaker of English in the field of the acoustic material. Simultaneously, the long sentences were broken down into several short sentences. Meanwhile, these modifications were highlighted in yellow in the revised manuscript.

  1. The results should be compared with the state-of-the-art related works.

Response:

Thank you very much for your significant suggestion. In the section 4.1.2, comparisons of the results obtained in this study with the state-of-the-art related works were added, which could exhibit the advances obtained in this research. Meanwhile, these modifications were highlighted in yellow in the revised manuscript.

  1. Why HR with the aperture for the cross sectional shape of round achieves the lowest resonance frequency? Please explain in detail.

Response:

Thank you very much for your kind suggestion. In the section 4.2.2, the major reason for the result that the HR with the aperture for the cross sectional shape of round achieves the lowest resonance frequency was analyzed and discussed. It could be observed from Figure 24 that the acoustic velocity for the HR with the aperture for round was larger than those for the HRs with other apertures, which could realize resonance in the lower sound frequency. Meanwhile, it could be observed that the sort of the acoustic velocity for the HR with various cross sectional shape from large to small in the Figure 12 was round, regular hexagon, square, regular triangle and regular pentagon, which was consistent with the simulation results in Figure 12. Meanwhile, these discussions were added and highlighted in yellow in the revised manuscript.

Reviewer 2 Report

The authors presented a very detailed and intersting investigation of the effects of of tangent and cross sectional shape on sound absorption for the acoustic metamaterial composed of Helmholtz resonators. At first they studied sound absorption mechanism for a single unit-cell using the finite element method. The results are validated experimentally and the authors showed a good agreement between numerical results and the experiment, so I strongly recommend the paper for publication in the present form.

The authors present original and very detailed study of the influence and aperture on the sound absorption coefficient, which is of practical interest.

The figures are of high quality, they are very neat and clear, and the only improvement is suggested: arrows in Fig. 2 showing radiuses could be added.

Discussion section has not been worked deep enough by the authors. It is key that authors discuss all the results here reported and compare their results with other findings previously reported by other researchers in the area.

Author Response

Response to reviewer 2

General Comment: The authors presented a very detailed and interesting investigation of the effects of tangent and cross sectional shape on sound absorption for the acoustic metamaterial composed of Helmholtz resonators. At first they studied sound absorption mechanism for a single unit-cell using the finite element method. The results are validated experimentally and the authors showed a good agreement between numerical results and the experiment, so I strongly recommend the paper for publication in the present form.

Response:

Thank you very much for your kind review to our manuscript and positive assessment to our research. We have revised the whole manuscript carefully according to your and other reviewers’ comments. The responses to your comments are as follows.

  1. The authors present original and very detailed study of the influence and aperture on the sound absorption coefficient, which is of practical interest.

The figures are of high quality, they are very neat and clear, and the only improvement is suggested: arrows in Fig. 2 showing radiuses could be added.

Response:

Thank you very much for your helpful suggestion. The arrows were added to the marked radiuses in the Figure 2. Meanwhile, the arrows were added to the marked parameters in the Figure 4 as well. These two figures were replaced in the revised manuscript.

  1. Discussion section has not been worked deep enough by the authors. It is key that authors discuss all the results here reported and compare their results with other findings previously reported by other researchers in the area.

Response:

Thank you very much for your significant suggestion. In the section 4.1.2, comparisons of the results obtained in this study with the state-of-the-art related works were added, which could exhibit the advances obtained in this research. Meanwhile, these modifications were highlighted in yellow in the revised manuscript.

Reviewer 3 Report

The paper is a numerical (with FEM) and experimental study to evaluate the sound absorption coefficient of Helmholtz resonators with different shapes of apertures. Although the parametric study and numerical results appear to be correct, there is no significant physical analysis of the results or important novelty in their work. The analysis of the numerical results is a mere description of the results of the simulations.

I recommend that the authors revise the paper to take these aspects into account.

- The title is unnecessarily long and contains subordinate clauses that are not keywords of the paper
- The abstract does not clearly show the main ideas of the paper, but very specific content with numerical values.
- The English is poor. The authors misuse long sentences, have many typos and make reading difficult overall.
- The quality of the figures can be improved. Especially with the 2D plots. Large legends with numerical values are not appropriate.

Author Response

Response to reviewer 3

General Comment: The paper is a numerical (with FEM) and experimental study to evaluate the sound absorption coefficient of Helmholtz resonators with different shapes of apertures. Although the parametric study and numerical results appear to be correct, there is no significant physical analysis of the results or important novelty in their work. The analysis of the numerical results is a mere description of the results of the simulations. I recommend that the authors revise the paper to take these aspects into account.

Response:

Thank you very much for your kind review to our manuscript and helpful assessment to our research. We have revised the whole manuscript carefully according to your and other reviewers’ comments. The responses to your comments are as follows.

  1. The title is unnecessarily long and contains subordinate clauses that are not keywords of the paper.

Response:

Thank you very much for your kind suggestion. The title is modified as “Effects of Aperture Shape on Absorption Property of Acoustic Metamaterial of Parallel–Connection Helmholtz Resonator”, which aims to reduce its length and make it concise.

  1. The abstract does not clearly show the main ideas of the paper, but very specific content with numerical values.

Response:

Thank you very much for your significant suggestion. The abstract is corrected according to your comment. Because the limit of the abstract by this journal ‘Materials’ is 200 words maximum, some content about numerical value is deleted and replaced by the main ideas of this study, and these modifications are highlighted in yellow in the revised manuscript.

  1. The English is poor. The authors misuse long sentences, have many typos and make reading difficult overall.

Response:

Thank you very much for your helpful suggestion. According to your and other reviewers’ comment, the language of whole manuscript was polished by a native speaker of English in the field of the acoustic material. Simultaneously, the long sentences were broken down into several short sentences. Meanwhile, these modifications were highlighted in yellow in the revised manuscript.

  1. The quality of the figures can be improved. Especially with the 2D plots. Large legends with numerical values are not appropriate.

Response:

Thank you very much for your helpful suggestion. According to your and other reviewers’ comments, the quality of the figures are improved, which aims to make this manuscript convenient to read, and the modifications are highlighted in yellow in revised manuscript.

Round 2

Reviewer 3 Report

Although the authors took great pains to correct some of the comments, they did not take into account the extensive revision and made only minor changes to the text and Figures.
Some of the comments have not been changed (legends with numbers ). The work is not worth publishing in its current form in the journal Materials. 

Author Response

Response to reviewer 3

General Comment: Although the authors took great pains to correct some of the comments, they did not take into account the extensive revision and made only minor changes to the text and Figures. Some of the comments have not been changed (legends with numbers). The work is not worth publishing in its current form in the journal Materials.

Response:

Thank you very much for your kind review to our revised manuscript and helpful comment to our research. We have revised the whole manuscript carefully according to your former comments in first round review. The responses to your comments are as follows.

  1. The title is unnecessarily long and contains subordinate clauses that are not keywords of the paper.

Response:

Thank you very much for your kind suggestion. The title is modified from original “Effects of Aperture Shape on Sound Absorption Performance of Acoustic Metamaterial of Parallel–Connection Helmholtz Resonator and Parameter Optimization” to “Effects of Aperture Shape on Absorption Property of Acoustic Metamaterial of Parallel–Connection Helmholtz Resonator”, which aims to reduce its length and make it concise. The modified title focuses on the “aperture shape”, “absorption property”, “acoustic metamaterial” and “Helmholtz resonator”, which are keywords of the paper.

  1. The abstract does not clearly show the main ideas of the paper, but very specific content with numerical values.

Response:

Thank you very much for your significant suggestion. The abstract is corrected according to your comment. Because the limit of the abstract by this journal ‘Materials’ is 200 words maximum, some content about numerical value is deleted and replaced by the main ideas of this study, and these modifications are highlighted in yellow in the revised manuscript.

In the abstract, the first and second sentences are “Helmholtz resonator (HR) with the embedded aperture is effective acoustic metamaterial for noise reduction in the low–frequency range. Its sound absorption property is significantly affected by the aperture shape.” They introduce the research background and focus of this research.

The third and fourth sentences are “Sound absorption properties of HRs with the embedded aperture for various tangent sectional shape were studied by two–dimensional acoustic finite element simulation. The sequence of resonance frequency from low to high was olive, common trapeziform, reverse trapeziform, dumbbell and rectangle.” They present the research method and main result for various tangent sectional shape.

The fifth and sixth sentences are “Meanwhile, those HRs for various cross sectional shape were investigated by three–dimensional acoustic finite element simulation. The sequence of resonance frequency from low to high were round, regular hexagon, square, regular triangle and regular pentagon.” They present the research method and main result for various cross sectional shape.

The seventh sentence is “Moreover, the reason for these phenomenons was analyzed by the distributions of sound pressure, acoustic velocity and temperature.” It presents the study on why the aperture with various shape exhibit different sound absorption property.

The eighth and ninth sentences are “Furthermore, on the basement of the optimum tangent and cross sectional shape, the sound absorption property of parallel–connection Helmholtz resonators was optimized. The experimental sample with optimal parameters was fabricated and its average sound absorption coefficient reached 0.7821 in the 500–820 Hz with the limited thickness 30 mm.” They present the experimental validation in this research, which includes parameter optimization and sample detection.

The tenth sentence is “The research achievements proved the significance of aperture shape, which provided guidance for the development of sound absorber in the low frequency range.” It present the research significance and application prospect of this study.

  1. The English is poor. The authors misuse long sentences, have many typos and make reading difficult overall.

Response:

Thank you very much for your helpful suggestion. In the first round revision, according to your and other reviewers’ comment, the language of whole manuscript was polished by a native speaker of English in the field of the acoustic material. Simultaneously, the long sentences were broken down into several short sentences.

In this second round revision, the whole manuscript is revised again to eliminate the typos and mistakes. Meanwhile, these modifications were highlighted in yellow in the revised manuscript.

  1. The quality of the figures can be improved. Especially with the 2D plots. Large legends with numerical values are not appropriate.

Response:

Thank you very much for your helpful suggestion. Maybe we didn't accurately understand your meaning in the first round revision. Sorry for our poor understanding. We don’t know the accurate meaning of “Large legends with numerical values are not appropriate”. The legend with same range in one figure are used to conduct the comparisons for different conditions. This time we enlarge the number in the legend, which aim to make it easier to read. If you feel the legend should not appear, we can delete them.

In this second round revision, the diagrammatic figures in the Figures 1, 2, 3, 4, 5, 6, 7, 8, 14, 15, 16 and 17 are improved their resolution ratios, which can make them clearer.

The data graphs in the Figures 9, 10, 11, 12, 13, 18, and 19 are improved their resolution ratios as well.

The marked legends in the Figures 20, 21, 22, 23, 24, 25 and 26 are adjusted. The size of numbers in the legend is enlarged, which aim to make it easier to read.
